# Space-time dependence of compound hot-dry events in the United States: assessment using a multi-site multi-variable weather generator

Manuela I. Brunner[1], Eric Gilleland[1], and Andrew W. Wood[1]

[1]Research Applications Laboratory, National Center for Atmospheric Research, 3450 Mitchell Ln, Boulder, CO 80301, USA

**Correspondence:** Manuela Brunner (manuela.i.brunner@gmail.com)

**Abstract.** Compound hot and dry events can lead to severe impacts whose severity may depend on their time scale and spatial extent. Despite their potential importance, the climatological characteristics of these joint events have received little attention regardless of growing interest in climate change impacts on compound events. Here, we ask how event time scale relates to (1) spatial patterns of compound hot-dry events in the United States, (2) the spatial extent of compound hot-dry events, and (3) the importance of temperature and precipitation as drivers of compound events. To study such rare spatial and multivariate events, we introduce a multi-site multi-variable weather generator (*PRSim.weather*), which enables generation of a large number of spatial multivariate hot-dry events. We show that the stochastic model realistically simulates distributional and temporal autocorrelation characteristics of temperature and precipitation at single sites, dependencies between the two variables, spatial correlation patterns, and spatial heat and meteorological drought indicators and their co-occurrence probabilities. The results of our compound event analysis demonstrate that (1) the Northwestern and Southeastern United States are most susceptible to compound hot-dry events independent of time scale and susceptibility decreases with increasing time scale, (2) the spatial extent and time scale of compound events are strongly related with sub-seasonal events (1–3 months) showing the largest spatial extents, and (3) the importance of temperature and precipitation as drivers of compound events varies with time scale where temperature is most important at short and precipitation at seasonal time scales. We conclude that time scale is an important factor to be considered in compound event assessments and suggest that climate change impact assessments should consider several instead of a single time scale when looking at future changes in compound event characteristics. The largest future changes may be expected for short compound events because of their strong relation to temperature.

## 1 Introduction

Compound hot and dry events, i.e. events that are extreme with respect to both temperature and precipitation, can lead to severe impacts in agriculture and other sectors as illustrated by the 2010 heatwave-drought in Russia or the 2012 heatwave-drought in the Central United States (US, Mo and Lettenmaier, 2015) which led to substantial reductions in crop yields (Wegren, 2011; Christian et al., 2020; Fuchs et al., 2012). The US has been shown to be affected by concurrent hot and dry events at different time scales including short and long events effective at weekly to monthly (Zhang et al., 2020) and seasonal to annual time scales (Alizadeh et al., 2020), respectively. The interest in these impactful compound events is reflected in an increasing number

of studies assessing changes in their frequency of occurrence. Substantial increases in the number of concurrent droughts and heat waves over the last few decades that are partly explained by increasing temperatures have been reported not just for the US (Alizadeh et al., 2020; Mazdiyasni and AghaKouchak, 2015; Tavakol et al., 2020) but also globally (Feng et al., 2020; Sarhadi et al., 2018) and for other regions of the world such as China (Wu et al., 2019; Zhou and Liu, 2018; Yu and Zhai, 2020) and Europe (Manning et al., 2019).

While frequency of occurrence is an important factor determining impacts, the severity of impacts related to compound events likely also depends on their spatial extent, i.e. how large the affected region is, and their time scale, i.e. whether they just last weeks or extend over a longer period of time. Indeed, the spatio-temporal behavior is a common target of analyses in general for drought, a related phenomenon, as in the multi-temporal severity-area-duration analyses presented by Andreadis et al. (2005). Despite their potential importance for understanding and projecting the physical manifestation and impacts of compound events, these spatio-temporal characteristics have received comparably little attention. Only recently, Alizadeh et al. (2020) and Wu et al. (2020) have shown that the area affected by concurrent hot-dry extremes has increased significantly over the past few decades in the US and globally for long, i.e. seasonal, time scales. However, it remains to be investigated how the time scale of compound events influences their characteristics and spatial extent.

This study aims to deepen our understanding of how the time scale of compound hot-dry events in the US relates to (1) spatial patterns of compound event affectedness (i.e. where in the US hot-dry events are most frequent), (2) spatial extents of compound events (i.e. how large compound events are), and (3) the role of temperature and precipitation as drivers of compound events by focusing on multivariate and spatial extreme events (Zscheischler et al., 2020). To answer the question of how time scale shapes compound event characteristics, we determine the probability, extent, and drivers of spatial multivariate heatwaves and meteorological drought over the conterminous US (CONUS) for different time scales ranging from weekly to annual events.

Studying such spatial multivariate events is challenging because they are rare in observational records (Zscheischler et al., 2018). This challenge can for example be tackled by developing stochastic simulation approaches to generate large data sets with similar statistical properties as the observations (Vogel and Stedinger, 1988). A stochastic approach to simulate spatial multivariate hot-dry events at different time scales needs to (1) represent spatial dependencies between sites to capture the spatial aspect, (2) represent dependencies between variables to capture dependencies between precipitation and temperature, and (3) be continuous to enable studying time scales from weeks to years. However, existing models often only fulfill one or two out of these three requirements. On the one hand, existing spatial models for simulating spatial extreme events such as the conditional exceedance model by Heffernan and Tawn (2004) are event-based (Keef et al., 2013; Diederen et al., 2019) and often applied to one variable, e.g. flood peaks. On the other hand, continuous stochastic approaches such as autoregressive moving average type models (Stedinger and Taylor, 1982) or bootstrap approaches (Rajagopalan et al., 2010) do not represent spatial dependencies well. Therefore, Brunner and Gilleland (2020) recently proposed a novel stochastic approach for simulating continuous streamflow time series in multiple catchments based on the wavelet transform. The Phase Randomization Simulation using wavelets (*PRSim.wave*) model combines an empirical spatio-temporal model based on the wavelet transform and phase randomization with the flexible four-parameter kappa distribution and builds on an earlier, univariate version of the

model (PRSim; Brunner et al., 2019). It is able to simulate continuous, spatially consistent time series but has so far only been applied to one variable (streamflow).

We extend *PRSim.wave*, here, to multiple variables by proposing a multi-site multi-variable stochastic weather generator (*PRSim.weather*) that simulates long time series of spatially consistent temperature ($T$) and precipitation ($P$) time series. This multi-site multi-variable stochastic model reproduces local variable distributions using flexible distributions for $T$ and $P$ and introduces spatio-temporal and variable dependence using the wavelet transform (Torrence and Compo, 1998) and phase randomization (Schreiber and Schmitz, 2000; Lancaster et al., 2018). Using this multi-site multi-variable generator to simulate a large set of spatial multivariate hot-dry events will help to shed light on the question of how time scale shapes compound event characteristics including spatial extent. Thus, this analysis will provide crucial information to increase preparedness and develop adaptation measures to potentially impactful spatial multivariate events.

## 2    Methods and Materials

We develop a multi-variable multi-site weather generator that stochastically simulates spatially consistent daily $T$ and $P$ time series for a large number of locations. We apply this model to a gridded $T$ and $P$ data set in the CONUS to generate a large sample of spatial multivariate hot-dry events. We subsequently use this sample to determine which regions in the US are susceptible to compound events and large spatial multivariate event extents at different time-scales. Last, we look at how the importance of $T$ and $P$ for compound event development varies with time scale.

### 2.1    Study region and data

The analysis is performed using a gridded data set of daily $T$ and $P$ time series for 894 equally-spaced grid cells in the CONUS. $T$ and $P$ data were obtained from the ERA5-Land reanalysis for the period 1981–2018 (ECMWF, 2019). ERA5-Land relies on atmospheric forcing from the ERA5 reanalysis (Hersbach et al., 2020) and provides variables at a spatial resolution of 9 km for the period 1981 to present. We chose a subset of regularly-spaced grid cells by sampling 1500 grid cells over the extent of the CONUS which resulted in 894 grid cells over land that are used for this analysis.

### 2.2    Methods

#### 2.2.1    Stochastic multi-site multi-variable modeling

To study compound hot-dry events, we develop a multi-site multi-variable weather generator *PRSim.weather* that enables simulation of large sets of spatially consistent compound hot-dry events at a daily scale. *PRSim.weather* combines an empirical spatio-temporal model based on the wavelet transform and phase randomization with two flexible parametric distributions for $T$ and $P$, which enables extrapolation to yet unobserved values. It builds on the spatial stochastic model *PRSim.wave* (Phase Randomization Simulation using wavelets) proposed by Brunner and Gilleland (2020), which simulates continuous streamflow

time series at multiple sites. We expand the functionality of *PRSim.wave* to simulate multiple variables, i.e. $T$ and $P$, at multiple sites. The weather generation procedure implemented in *PRSim.weather* consists of five main steps (Figure 1):

1. feed in observed daily $T$ and $P$ time series for multiple sites (here grid cells);

2. fit monthly distributions to $T$ and $P$ time series at each site to capture seasonal variations in distribution parameters (i.e. one separate distribution is fitted to the data in each month). Using theoretical instead of empirical distributions will allow us to generate extreme values more extreme than the observations. For $T$, we use the flexible skewed exponential power (SEP) distribution with 4 parameters (Fernández and Steel, 1998), which generalizes the Gaussian distribution, can reproduce different skewness and kurtosis, and has been previously applied for multi-site temperature simulation (Evin et al., 2019). The SEP distribution is defined as

$$F(x) = \begin{cases} [\kappa^2/(1+\kappa^2)]\gamma\{[(\xi-x)/(\alpha\kappa)]^h, 1/h\} & \text{for } x < \xi \\ 1 - [1/(1+\kappa^2)]\gamma\{[\kappa(x-\xi)/\alpha]^h, 1/h\} & \text{for } x \geq \xi, \end{cases} \tag{1}$$

with location parameter $\xi$, scale parameter $\alpha$, shape parameters $\kappa$ and $h$, and $\gamma(Z, \alpha)$ representing the upper tail of the incomplete gamma function (Asquith, 2014).

The parameters of the SEP distribution are estimated using L-moments (R-package *lmomco*; Asquith, 2020). For $P$, we use an extended Generalized Pareto distribution (E-GPD; Papastathopoulos and Tawn, 2013) with three parameters to model positive precipitation values. The E-GPD jointly models non-extreme and extreme values of $P$ while bypassing the threshold selection problem as it enables smooth transitioning between a gamma-like distribution and a heavy-tailed Generalized Pareto distribution (GPD) thanks to a transformation function $G(v)$ (Naveau et al., 2016). The E-GPD is defined as

$$F\{x\} = G[H_\theta\{x/\sigma\}], \qquad \text{where} \qquad H_\theta(z) = \begin{cases} 1 - (1+\theta z)^{-1/\theta} & \text{if } \theta \neq 0, \\ 1 - e^{-z} & \text{if } \theta = 0, \end{cases} \tag{2}$$

where $\sigma > 0$ is a scale parameter, $\theta$ is the shape parameter of the GPD, and $G(v) = v^\rho$. The E-GPD has been demonstrated to be valuable in multi-site precipitation modeling thanks to its flexibility (Evin et al., 2018). The parameters of the E-GPD distribution are estimated using probability weighted moments (R-package *mev*; Belzile et al., 2020). We use the E-GPD to simulate non-zero precipitation values and complement it with as many zero-values as in the observations to obtain the full $P$ distribution with appropriate probability of precipitation occurrence;

3. transform the $T$ and $P$ time series from the time to the frequency domain by decomposing the series into an amplitude and phase signal using a continuous wavelet transform with the Morlet wavelet (Torrence and Compo, 1998) (R-package wavScalogram; Bolós and Benítez (2020)). The continuous wavelet transform is defined as the convolution of a time series $x_n$ of length $n$:

$$W_n(l) = \sum_{n'=0}^{N-1} x_{n'} \psi_0^* \left[ \frac{(n'-n)\delta t}{l} \right], \tag{3}$$

where the (*) indicates the complex conjugate, $l$ the wavelet scale, and $\psi_0(\eta)$ the Morlet wavelet, which is defined as

$$\psi_0(\eta) = \pi^{-1/4} e^{i\omega_0\eta} e^{-\eta^2/2}, \tag{4}$$

where $\eta$ is a non-dimensional time parameter, $\omega_0$ is the non-dimensional frequency, and $i = \sqrt{-1}$ is the imaginary unit;

4. generate one random time series using bootstrap resampling on the temperature time series of one randomly sampled site by sampling years with replacement. Use the wavelet transform to also decompose this bootstrapped series in order to obtain a random phase signal;

5. generate stochastic time series for $T$ and $P$ by applying the inverse wavelet transform to the observed amplitude signals and the randomly generated phases. Rank-transform the newly generated time series using the probability integral transform to the desired distribution for each month using the monthly distribution parameters derived in Step 2 (SEP parameters for $T$ and E-GPD parameters for $P$).

The simulation of yet unobserved magnitudes becomes possible thanks to the use of parametric distributions for $T$ and $P$ in Step 2. The spatial and variable dependencies are introduced in Step 5 by using the same random phases in the wavelet transform at all sites and for both variables.

The stochastic multi-site multi-variable model is evaluated with respect to the following characteristics: (1) $T$ and $P$ distributions (cdfs) at individual sites, (2) temporal autocorrelation of $T$ and $P$ (acfs) at individual sites, (3) spatial dependencies across sites for $T$ and $P$ (variograms), (4) $T$-$P$ variable dependencies (scatter plots), and (5) simulated spatial patterns of the standardized temperature index (STI), the standardized precipitation index (SPI), and the probability of compound high STI and low SPI anomalies at a 1-month aggregation level for moderate, severe and extreme events according to the empirical copula (see Section 2.2.2).

*PRSim.weather* is finally run $n = 100$ times for the 894 grid cells in the US in order to substantially increase the sample size available for the assessment of compound hot-dry events by pooling the different model runs (28 years * 100 = 2800 years).

### 2.2.2 Compound event analysis

While the focus is on the simulated series, compound events and their corresponding $T$ and $P$ characteristics are identified at different time scales in both the observed and stochastically simulated time series to assess the reliability of the stochastic model. To look at different time scales, we first convert the $T$ and $P$ series to weekly/monthly series using mean values and sums, respectively. We work with aggregation levels of 1 week to represent 'flash' compound events and of 1, 3, 6, and 12 months to represent sub-seasonal, seasonal, and annual time scales. In a second step, we transform the aggregated $T$ and $P$ series to series of standardized indices, which we will use to study relationships between the marginal behavior of compound events because they guarantee variable and site comparability. Standardized precipitation index (SPI) series (McKee et al., 1993) for each location are computed by transforming the $P$ values to a standardized normal distribution (mean = 0 and sd = 1) using a site-specific E-GPD distribution (Kolmogorov–Smirnov test did not not reject Gamma in over 80% of the grid cells). Similarly, we compute standardized temperature index series (STI; Zscheischler et al., 2014) using the SEP distribution

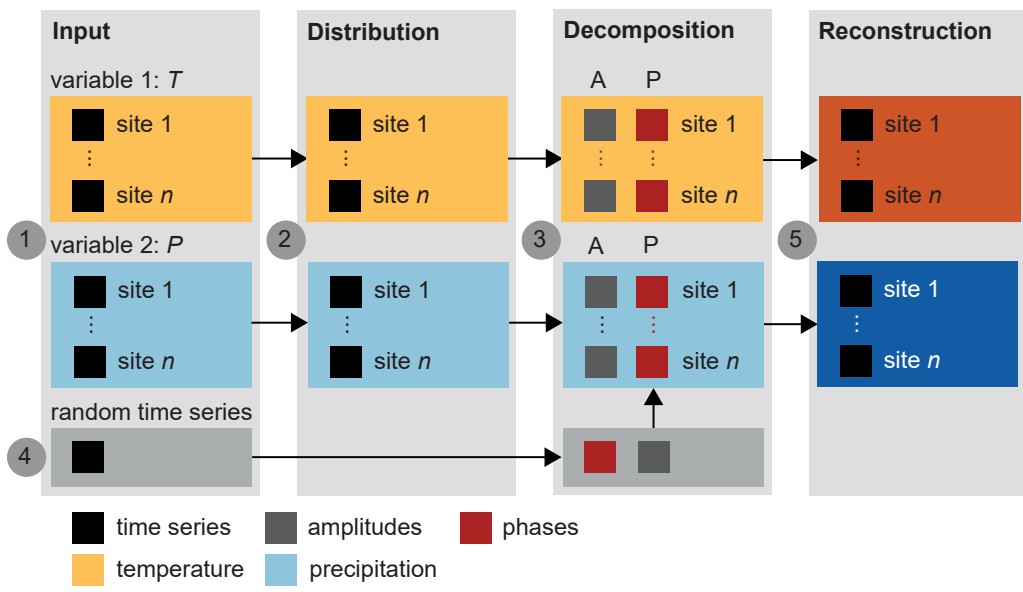

**Figure 1.** Illustration of 5 working steps of *PRSim.weather*: (1) Feed in daily observed temperature ($T$) and precipitation ($P$) time series for multiple sites $(1, ..., n)$; (2) fit SEP distribution to $T$ and E-GP distribution to $P$ time series of all sites at a monthly scale; (3) decompose $T$ and $P$ time series of all sites into an amplitude (A) and phase (P) signal using the wavelet transform; (4) generate one random time series using bootstrap resampling and decompose that random series into an amplitude and phase signal too, and (5) generate random daily $T$ and $P$ time series by combining the observed amplitude signal of each site and variable with the randomly generated phase signal and by backtransforming the signals to the time domain using the inverse wavelet transform. Rank-transform the newly generated signal to the desired distribution using the parameter estimates from Step 2.

for transformation. Last, compound hot-dry events are identified for each time scale and grid cell using a bivariate empirical copula (Deheuvels, 1979; Genest and Favre, 2007), which describes the joint distribution of $T$ (STI) and $P$ (-SPI) with uniform margins. We change the sign of the SPI values to convert negative to positive anomalies as we are interested in events where STI and -SPI are extreme. The empirical copula of STI and -SPI is described as:

$$C_n(u,v) = \frac{1}{n} \sum_{i=1}^{n} 1\Big(\frac{R_i}{n+1} \leq u, \frac{S_i}{n+1} \leq v\Big), \tag{5}$$

where $R_i$ and $S_i$ represent pairs of ranks (across STI and SPI time series), $n$ the sample size, and $C_n(u,v)$ the rank-based estimator of the copula $C(u,v)$. An example of how the empirical copula (purple) is related to the margins STI (yellow) and -SPI (blue) is provided in Figure 2.

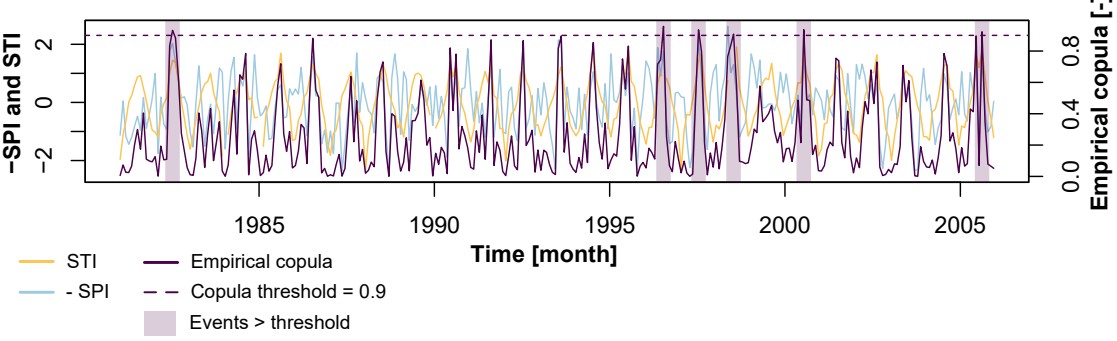

**Figure 2.** Illustration of the relationship between monthly STI (yellow) and -SPI (bue) time series and their bivariate copula (i.e. the values of $C_n\left(\frac{R_i}{n+1}, \frac{S_i}{n+1}\right)$; purple)

for one example grid cell. Compound STI and -SPI events exceeding a copula-threshold of 0.9 are highlighted by purple boxes.

Using the time series of empirical bivariate distribution values, we identify moderate, severe, and extreme compound events using three thresholds at 0.8, 0.9, and 0.95, respectively (see Figure 2 for an example with a threshold of 0.9). This copula-based threshold procedure slightly differs from an approach where both margins (-SPI and STI) have to jointly exceed a threshold in order for an event to be defined as a compound event. The bivariate threshold procedure includes a slightly different event space, which besides the jointly marginally extreme events also includes those events that are extreme in terms of the bivariate distribution but not necessarily in terms of both margins. Please note that the focus on high $T$ and low $P$ events leads to the selection of compound events in the summer season. For an aggregation period of 1 month, all selected compound events happen between May and October with over 90% of the events happening in July or August. The seasonal focus is slightly shifted towards late summer (august) and early fall (september and october) as we move towards longer aggregation periods.

To assess the spatial extent of compound events at different time scales, we define the spatial extent of the compound event as the percentage of grid cells affected by the compound event at any given time scale. Then, for each grid cell, we determine the median spatial extent of those events it is affected by at each time scale.

To explain the role of the individual variables $T$ and $P$ in compound event occurrence, we compute Kendall's correlation between the median bivariate distribution (empirical copula) and the median standardized indices STI and SPI over all simulation runs at different time scales. This correlation analysis is performed for nine hydro-climatic regions in the United States (Bukovsky; Bukovsky, 2011) to quantify the regional spread in the role of STI and SPI for compound event development, i.e., correlation is computed between median bivariate distributions and median STI or SPI at different grid cells within a region. We look at correlations for different time scales and event extremeness levels to assess to which degree these two factors influence STI and SPI importance.

# 3 Results

## 3.1 Evaluating the weather generator

The multi-site multi-variable stochastic simulation approach *PRSim.weather* is capable of reproducing the observed statistical characteristics of $T$ and $P$ time series at individual locations as illustrated by one example station (Figure 3). The flexible SEP and E-GPD distributions capture the local $T$ and $P$ distributions well as indicated by the good match of simulated with observed densities (Figure 3a,b). The suitability of the SEP and E-GPD distributions to model local $T$ and $P$ distributions also extends to the tails as 100-year return levels estimated from the observed and simulated series compare well for both variables. The temporal autocorrelation in both variables is realistically reproduced as shown by the good agreement of simulated with observed autocorrelation functions thanks to the observed frequency spectrum information used in the inverse wavelet transform (Figure 3c,d). The simulated time series well mimick the main temporal characteristics of the observed time series including seasonality and temporal event distribution/clustering as illustrated by three years of observed and simulated $T$ and $P$ data (Figure 3e,f). The $T–P$ variable dependence is also generally well captured thanks to the use of the same random phases for both variables when applying the inverse wavelet transform (Figure 3g,h). However, the number of high $T$–low $P$ events at a daily scale is slightly underestimated. The above-described model evaluation can be generalized to other grid cells in the data set. In addition to these local characteristics, spatial correlations are captured as illustrated by the similarity of observed and simulated variograms (Figure 4). However, the spatial correlation of $T$ is slightly overestimated by the simulations. Achieving a 'perfect' joint representation of the three forms of dependence – temporal, spatial, and variable – is very challenging. The model is considered suitable for the analysis of compound hot-dry events because it has an acceptable performance with respect to all three aspects and enables increasing the sample size of compound events.

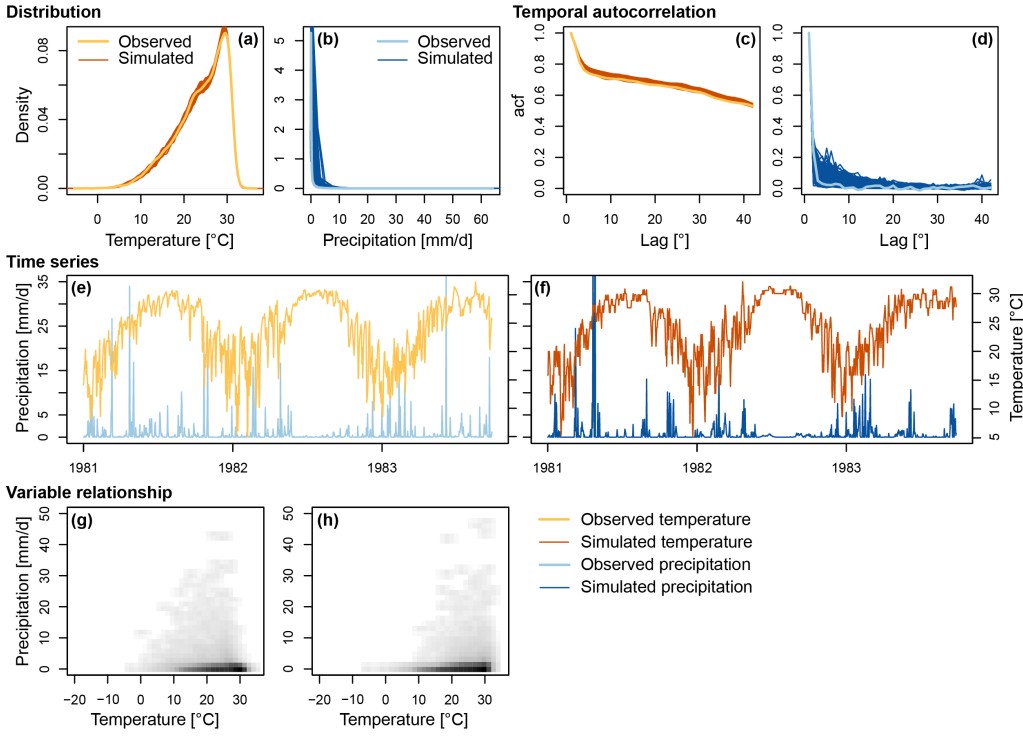

**Figure 3.** *PRSim.weather* evaluation for one example grid cell: (a, b) marginal distributions for observed and simulated $T$ (orange) and $P$ (blue) (one line represents one simulation run), (c, d) temporal autocorrelation for observed and simulated $T$ and $P$ (one line represents one simulation run), (e, f) 3-year time series of $T$ (right y-axis) and $P$ (left y-axis) for observations and simulations, and (g, h) heat scatter plot of $P$–$T$ relationship in observations and simulations.

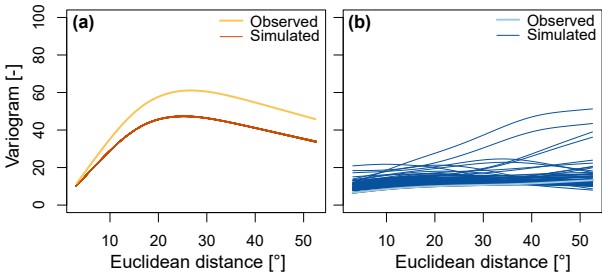

**Figure 4.** *PRSim.weather* evaluation for spatial dependence: (a) observed vs. simulated $T$ (orange) variograms for 92 equally spaced grid cells and (b) observed vs. simulated $P$ (blue) variograms for 92 grid cells, which describe the degree of spatial dependence of a field (Cressie, 1993).

*PRSim.weather* enables simulation of a large sample of extreme events in terms of standardized temperature (STI) and precipitation indices (SPI). These spatial samples enable comparing observed and simulated STI and SPI patterns for different levels of extremeness (Figure 5). While the simulated spatial STI and SPI patterns look similar as the observed ones, they

are more expressed because of the larger sample available, which contains yet unobserved extremes because of the use of parametric distributions for simulating $T$ and $P$. The spatial pattern for STI is rather weak with STI values being relatively homogeneously distributed except for the Pacific Northwest and along the West Coast where STI values are slightly higher than in the rest of the country. In contrast, the spatial pattern of median SPIs is expressed with substantially higher negative anomalies in the western than the eastern US and particularly strong negative anomalies in the Southwest.

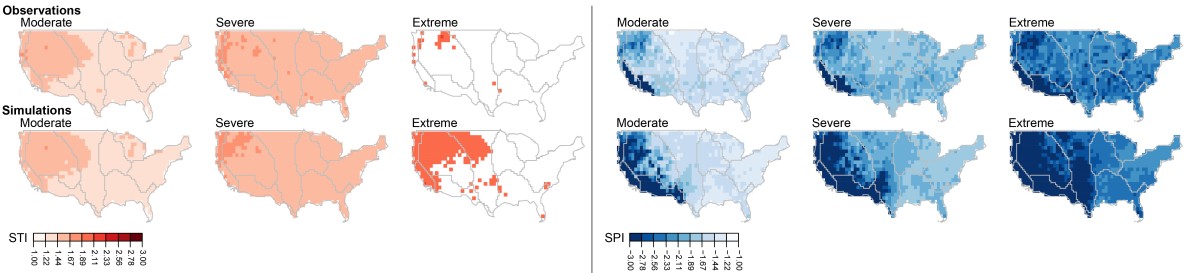

**Figure 5.** Spatial distribution of median observed (upper panel) and simulated (lower panel) STIs (left panel) and SPIs (right panel) at a monthly time scale for three levels of extremeness: moderate (STI > 1 and SPI < -1), severe (STI > 1.5 and SPI < -1.5) and extreme (STI > 2 and SPI < -2). The darker the color, the more severe are median events in a certain grid cell.

The spatial STI and SPI patterns are reflected in the spatial distribution of the probability of compound hot-dry events, which is also realistically represented but slightly underestimated by *PRSim.weather* (Figure 6). The highest probability of compound hot-dry events at a monthly time scale are found in the Pacific Northwest, along the West coast, in the Rocky Mountains, and the Southeast, in particular in Texas. In contrast, compound hot-dry events are relatively rare in the Great Plains, the Midwest, and Florida. For the remainder of our analysis, we are focusing on the stochastic simulations because of their large sample size, which allows us to study rare spatial multivariate hot-dry events.

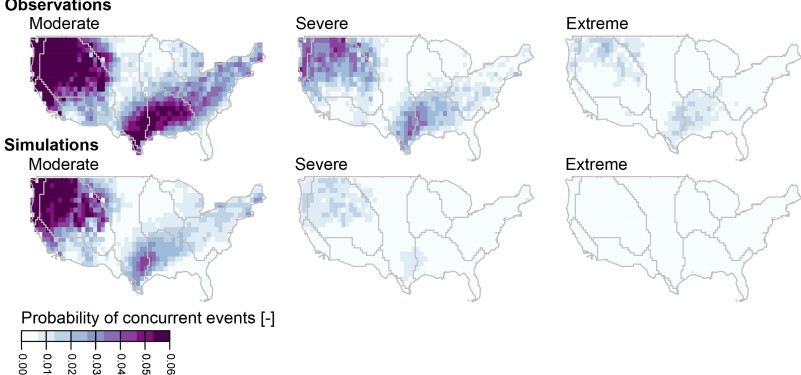

**Figure 6.** Spatial distribution of observed (upper panel) and simulated (lower panel) probability of occurrence of hot-dry events (number of compound events/total number of months, -) at a monthly time scale for three levels of extremeness: moderate ($C_n > 0.8$), severe ($C_n > 0.9$), and extreme ($C_n : 0.95$). The darker the color, the more likely are compound hot-dry events.

## 3.2 Compound hot-dry events

The stochastically simulated compound hot-dry events reveal that the probability of co-occurring hot and dry periods is highest in the northwestern and southeastern US independently of the time scale considered (Figure 7). However, the probability of compound events decreases with increasing duration, as can be expected due to the aggregation over increasingly longer periods of multiple weather events that may not all favor instantaneous compound hot and dry conditions, and joint extremeness. Still, there are spatial nuances depending on the time scale considered. For example, the high probabilities of compound events are located in the south for short time scales and move to the southeast as we move towards longer time scales. Time scale does not only affect local concurrence probabilities but also the size of the regions affected by compound hot-dry events, which decreases with increasing time scale and event extremeness. At an annual time scale, the probability of events at all extreme thresholds is negligible.

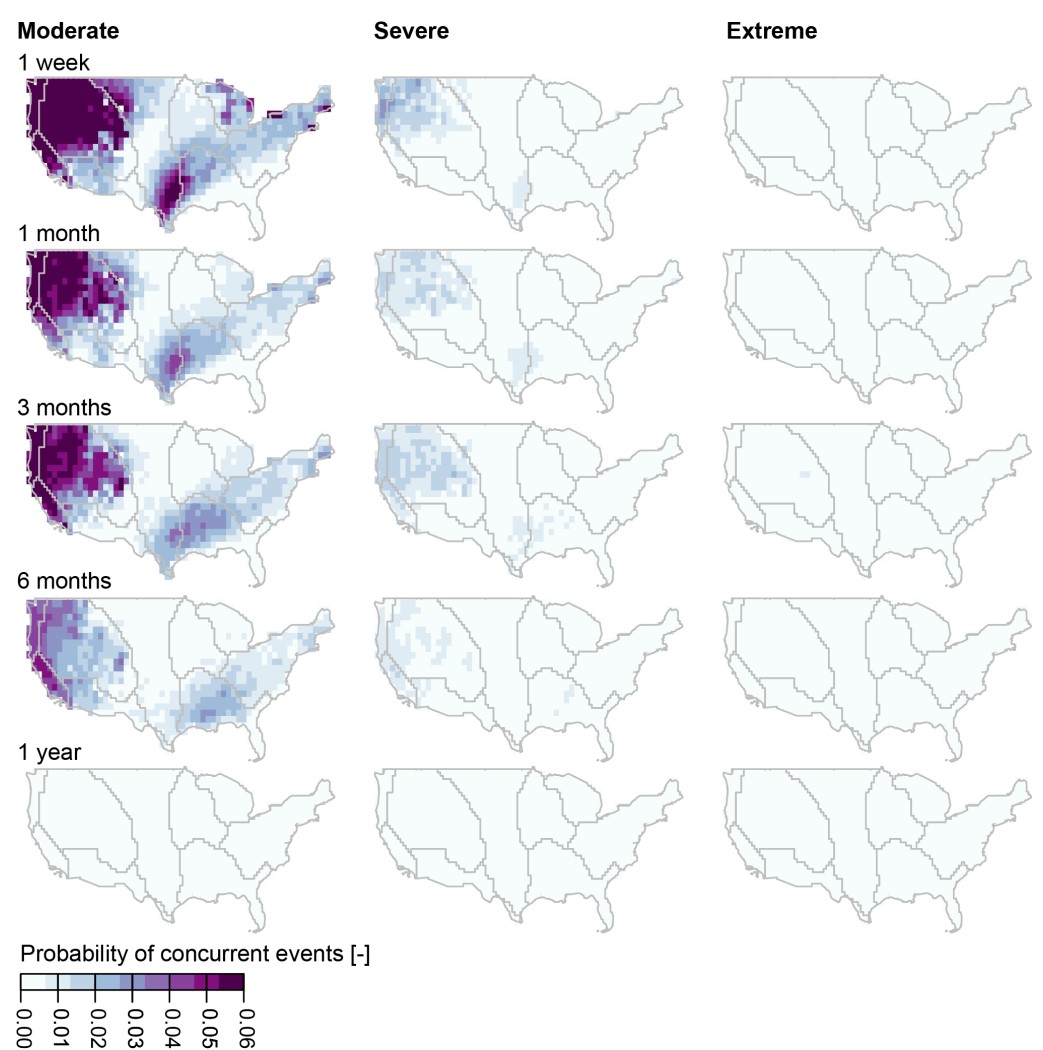

**Figure 7.** Probability of compound hot-dry events (number of compound events/total number of months, -) at different time scales (1 week, 1 month, 3 months, 6 months, 1 year) and for three levels of extremeness (moderate $C_n > 0.8$, severe $C_n > 0.9$, and extreme $C_n > 0.95$) per grid cell. The darker the color, the higher the probability that a grid cell is affected by compound hot-dry events.

Different regions of the US differ not only in how susceptible they are to compound hot-dry event occurrence but also in how likely they are to be affected by a widespread (large spatial scale) compound event. The spatial occurrence patterns for spatially extensive compound hot-dry events vary by time scale (Figure 8). For moderate extremes, the Midwest is most affected by large events, with more prevalence in the upper Midwest at shorter time scales and the central to southern Midwest at longer time scales. For the severe category, the western and Southeastern regions are more affected, a similar spatial pattern to the

probability of compound hot-dry events (Figure 7), although there are no large-scale events at short time scale. In addition, large compound events generally become less likely as we move beyond the 3 month time scale and toward extreme events

(Figure 9). While ~20% of the CONUS may be jointly affected by moderate and short compound events, spatial extents of compound events become small to non-existent for extreme and long-lasting (i.e. annual) compound events.

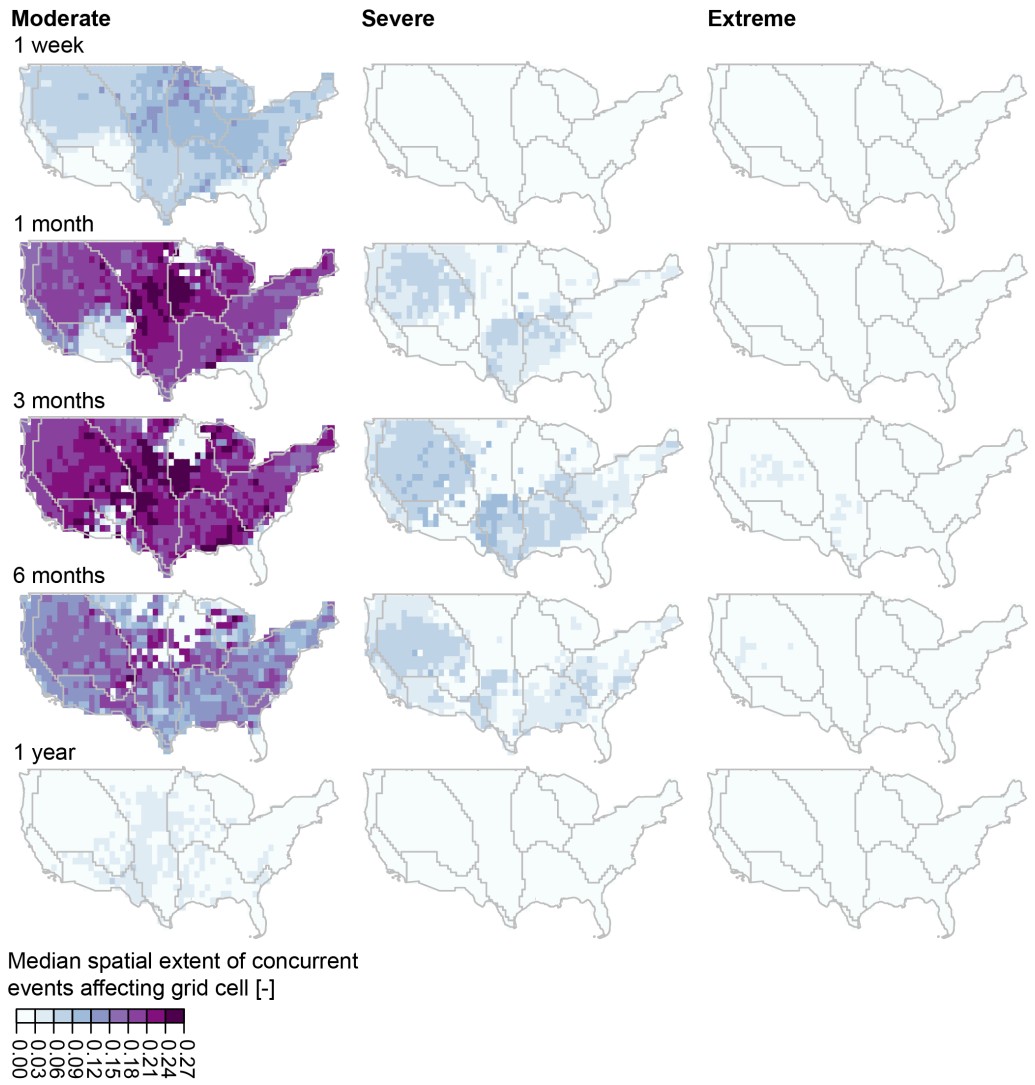

**Figure 8.** Spatial patterns of median compound event extent per grid cell for different time scales and extremeness levels over nine hydro-climatic regions. The darker the color, the higher the median spatial extent of compound events a grid cell is affected by.

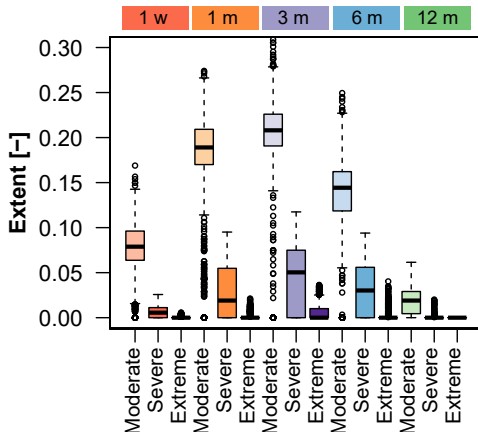

**Figure 9.** Relationship between compound event extent, time scale, and event extremeness. Boxplots summarize the spread of median extent (percentage of overall area affected, -) across grid cells for weekly (red), monthly (orange), 3-monthly (purple), 6-monthly (blue), and annual (green) time scales and three levels of extremeness: moderate, severe, and extreme.

The importance of $T$ (STI) and $P$ (SPI) as drivers of compound events varies by time scale and level of extremeness (Figure 230 10). $T$ is a particularly important driver at short time scales as indicated by the high correlation between median STI and median bivariate distribution of grid cells within a specific hydro-climatic region (Figure 10a). The importance of $P$ as a driver of compound events increases with time scale up to event durations of 6 months but decreases with level of extremeness (Figure 10b). In summary, the longer the time scale, the more important $P$ becomes as a driver compared to $T$ (up to a seasonal time scale).

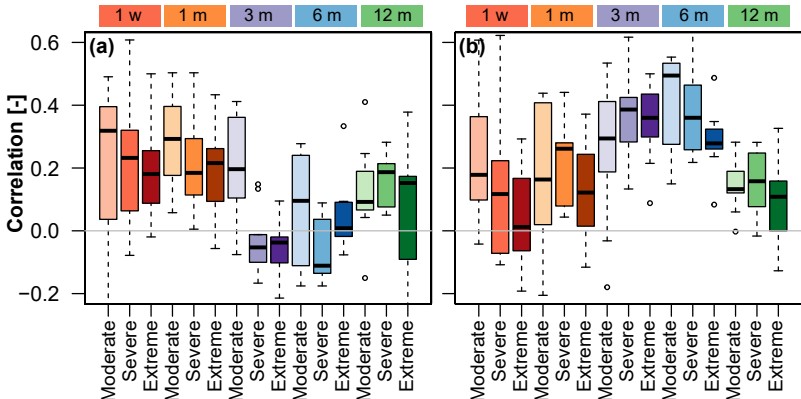

**Figure 10.** Importance of $T$ and $P$ as drivers of compound events across time scales and extremeness levels. Correlation of median bivariate distribution (empirical copula) per grid cell with median (a) STI and (b) SPI per grid cell cell. Correlations were computed using all simulation runs for nine hydro-climatic (Bukovsky) regions (spread of boxplot) per time scale (color) and level of extremeness (hue).

## 4  Discussion

The multi-site multi-variable stochastic model *PRSim.weather* proposed for the joint simulation of $T$ and $P$ at multiple sites has been shown to be suitable for the simulation of spatial multivariate hot-dry events. It reproduces the distributional and temporal autocorrelation characteristics of $T$ and $P$ at single sites, the dependence between the two variables, the spatial correlation of $T$ and $P$ across sites, and spatial patterns of STI, SPI, and their concurrence probabilities. However, spatial dependencies are slightly overestimated while variable dependencies are slightly underestimated. The model still has acceptable performance across three types of dependencies – temporal, spatial, and variable – and enables studying rare spatial multivariate events, which would not be possible using observations only. Please note that even though the model generates yet unobserved observations, the simulations are not independent of the limited sample size used to fit the model because the model is data-driven as any other calibrated/fitted model. Please also note that while the model will be able to retain the statistical dependencies between variables to some degree, individual simulated events may not necessarily be physically consistent if many variables are jointly simulated. We note that stochastic approaches may be combined with physical approaches as e.g. in the weather generator AWE-GEN-2 by Peleg et al. (2017) or one may rely on large climate ensemble simulation approaches (Deser et al., 2020; Bevacqua et al., 2021).

Further model development should focus on how to improve the representation of dependencies in very high $T$–low $P$ events at a daily scale and applications in other contexts and under non-stationary conditions. While the current application focuses on the two variables $T$ and $P$ in the US, the model can be adapted to other regions, other variables, and a multivariate context where more than two variables are of interest. Adapting the model to other regions and variables requires reconsidering distribution choices and extending it to a multivariate context necessitates adding more input variables, which are subsequently randomized in the same way as all other variables. Potential multivariate applications include the simulation of spatial concurrent pluvial, river, and coastal flooding by jointly modeling precipitation, discharge, and water levels or the joint simulation of wildfire drivers such as wind speed, temperature, and humidity. Extending model application to non-stationary conditions would require the implementation of non-stationary distributions for both $T$ and $P$. For example, one could introduce covariates for certain parameters of the marginal distributions of $T$ and $P$ in Step 2 or introduce covariates with information about trends or variability in $P$ and/or $T$ to guide resampling in Step 4.

The finding that the western and southeastern US are most likely to be affected by compound hot-dry events at sub-annual time scales suggests that the likelihood of compound events is somehow related to precipitation seasonality with regions receiving most of their precipitation in winter or spring and comparably less in summer and fall (Finkelstein and Truppi, 1991) being most likely to be affected by compound events. In 'normal' years, both the western and southeastern US receive a large part of their precipitation through recurrent patterns such as atmospheric rivers (Rutz et al., 2015) or tropical cyclones (Kunkel et al., 2012), respectively. Anomalies can arise because of temporal shifts or a weakening of these patterns in specific seasons/years. In addition, the regions most likely to experience compound events are the regions found to be most susceptible to heatwaves in the US (Smith et al., 2013).

Our finding that spatial extents of compound events are largest for moderate events at subseasonal time scales implies that while these moderate events may have less severe impacts at a local scale, they may still be highly relevant at a regional scale. Compound events with large spatial extents represent a particular management challenge because they may preclude the transfer of resources and emergency supplies from one to another region. Consequently, the societal impacts of large-scale compound events can be amplified, since many coping strategies are predicated on some degree of resource transfers from less severely affected adjacent regions (Murgatroyd and Hall, 2020).

The finding that temperature is a comparably more important driver for short compound events only while precipitation is comparably more important at seasonal time scales corroborates the findings of previous studies about the importance of different hydro-meteorological drivers at different time scales. Zhang et al. (2020) have shown that temperature is the most important hydro-meteorological driver of short term compound hot-dry extremes, which aligns with our findings. In addition, Tavakol et al. (2020) have shown that at long (i.e. annual) time scales, hot-dry-windy events co-occurred with major heat waves, which is in line with our finding that temperature is an important driver of extreme compound hot-dry events at seasonal to annual time scales.

Future changes in the frequency and severity of compound hot-dry events are expected because of both changes in temperature and precipitation and their interdependence. The importance of temperature as an important driver of short and extreme compound hot-dry events suggests that the increasing temperatures associated with climate change may induce future changes in the frequency and magnitude of short and extreme compound events. Such future increases have been projected globally (Wu et al., 2020) and regionally e.g. for China (Zhou and Liu, 2018). In addition, previous studies have shown that the number and intensity of compound hot-dry events may increase because temperature and precipitation may become increasingly coupled/correlated in summer (De Luca et al., 2020; Zscheischler and Seneviratne, 2017) possibly as a consequence of an intensification of land-atmosphere feedbacks (Seneviratne et al., 2010). As the number of compound events increases locally, the area exposed to compound hot-dry events is projected to increase with global warming (Vogel et al., 2019) continuing a trend that has been already observed during the past few decades (Alizadeh et al., 2020). How exactly future changes in compound event extents relate to changes in drought spatial extent (Brunner et al., 2021) and in heatwave spatial extent remains to be investigated.

## 5 Summary and Conclusions

We introduce the multi-variable multi-site stochastic model *PRSim.weather* to simulate continuous and spatially consistent multivariate time series. The model is shown to realistically simulate distributional and temporal autocorrelation characteristics of temperature and precipitation at single sites, dependencies between the two variables up to moderate extremes, spatial correlation patterns, and spatial heat and drought indicators and their co-occurrence probabilities for a gridded large-sample data set in the United States. However, future work is needed to improve the representation of very extreme hot-dry events. We apply the stochastic model to generate a large set of spatial and multivariate hot-dry events and use these simulated compound events to assess how event time scale and extremeness influence the spatial affectedness by compound hot-dry events over the

United States, the spatial extent of compound events, and their main drivers temperature and precipitation. Our results show that (1) the Northwest and Southeast are most likely to be affected by compound hot-dry events independent of time scale; (2) the spatial extent of compound hot-dry events decreases with increasing event extremeness and time scale, i.e., the events with the largest spatial extents are typically short and only moderately extreme; and (3) temperature is an important driver of short compound events while precipitation is an important driver at seasonal time scales particularly for the moderately extreme events. These findings highlight that occurrences of compound events are strongly influenced by the time scales at which they are defined. Research to quantify current compound event risk and to project it into the future will need to take time scale into consideration, especially as it also influences the sensitivity to different climate drivers and their potential future changes. Considering space-time scales in compound event assessments will allow us to make nuanced statements about which types of compound events may be changing because of increasing temperatures in a warming world. For example, short compound events and therefore events with large spatial extents may become more frequent with increasing temperatures, which will pose new regional management challenges.

*Code and data availability.* The ERA5-Land temperature and precipitation data used for this analysis can be downloaded from the Copernicus Climate Data Store: https://cds.climate.copernicus.eu/cdsapp#!/dataset. The stochastic weather generator *PRSim.weather* is implemented in the R-package *PRSim* under the function *PRSim.weather* and available for download: https://cran.r-project.org/web/packages/PRSim/.

*Author contributions.* MIB developed the study concept and stochastic simulation model, performed all the analyses, and wrote the first draft of the manuscript. EG provided methodological advice and revised and edited the manuscript. AW contributed to the interpretation of the results and revised and edited the manuscript.

*Competing interests.* The authors declare no competing interests.

*Acknowledgements.* This work was supported by the Swiss National Science Foundation via a PostDoc.Mobility grant (Number: P400P2_183844, granted to MIB). We would like to acknowledge high-performance computing support from Cheyenne (doi:10.5065/D6RX99HX) provided by NCAR's Computational and Information Systems Laboratory, sponsored by the National Science Foundation.

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
