# Peer review of "Space-time dependence of compound hot-dry events in the United States: assessment using a multi-site multi-variable weather generator"

_Earth System Dynamics, 2021_

## Referee Comment (RC1)

Review of the manuscript "Space-time dependence of compound hot-dry events in the United States: assessment using a multi-site multi-variable weather generator" by Manuela I. Brunner, Eric Gilleland, and Andrew W. Wood.

**General comment**

The authors introduce a multi-site multi-variable weather generator (*PRSim.weather*), which allows for simulating Temperature and Precipitation over the US during 100*28 years. While the weather generator has some limitations (that the authors discuss), the output is overall satisfying. The simulated data allows for analysing both (1) events that tend to be characterised by hot and dry conditions and (2) the spatial extents of these events. The authors illustrate and discuss the characteristics of these events across the US. Some improvements is needed, especially in the presentation of some methodological aspect (selection of concurrent hot and dry events and method for analysis in Figure 10). Generally, the paper is definitely well structured and I found it interesting.

I recommend the authors to consider my specific comments below. Those marked with *** are the less technical.

**Specific comments**

L9 **meteorological** drought indicators

L25. Could you mention, very *briefly* as it is an introduction, what were the causes for the changes in hot and dry events in these studies, e.g., temperature/precipitation trends?

L30 I suggest re-shaping the sentence slightly. That is, including the words "local" and "regional" (or "aggregated over a region"). The local impact depends on frequency and duration. The aggregated regional impacts depend, in addition, also on the extent.

*** L40. I think that the terminology could be improved, not only here, despite it is not wrong as there is no full agreement on this matter in the community. For example, here: "(2) spatial extents of compound events".

- You use "compound event" to refer to concurrent events or multivariate events (such as hot-dry events), which is a type of compound event.

- Note that the spatial characteristics of an event, make the event compound on its own (Zscheischler et al., 2020).

Therefore, the considered events are compound for two reasons, however you refer to the multivariate characteristic as a compound element, but you do not do the same for the spatial part. Of course, you cannot say twice "compound", but why is one privileged? Talking of "spatial extents of concurrent hot and dry events" may make things better in the paper. This would lead to reshaping a bit, for example, lines 40-45.

L40. Spatial patterns and spatial extents. Please, make the difference clear. I know what you mean, but I suspect that it will not be obvious to everyone.

L45. This statement is interesting. We have recently worked on the topic and shown that it is very

difficult to study seasonal precipitation extreme extents without large ensemble simulations (discussed at the end of the "Present-day spatial scale extremes" section):

*Bevacqua, E., Shepherd, T.G., Watson, P.A.G., Sparrow, S., Wallom, D., and Mitchell, D. (2020). "Larger spatial footprint of wintertime total precipitation extremes in a warmer climate". Submitted. Preprint's DOI: 10.1002/essoar.10505310.1*

\*\*\* L45 You write: "Studying such spatial compound events is challenging because they are rare in observational records (Zscheischler et al., 2018). This challenge can be tackled by developing stochastic simulation approaches to generate large data sets with similar statistical properties as the observations "

The weather generator is calibrated on and learn from the limited observation (or available data). So, does using a weather generator address completely the challenge of limited data? I suggest discussing this, especially the limitations, for a non-expert reader.

L76 add "daily" to "time series".

L83 Also in the procedure. You simulate, in the end, daily time series of P and T. Could you state this explicitly somewhere, maybe simply adding a "daily" somewhere?

Caption Fig 1. Add "daily" and "monthly" where required. E.g., in Step (2), I suggest moving the "monthly": "fit SEP distribution to T and E-GP distribution to P **monthly** time series of all sites"

L119 Adding "aggregated" somewhere may help to make very clear that you will pull together all the weather generator output in a unique aggregated time series of 2800 years (one may in principle repeat the analysis on the 100 weather generator output and get, e.g., a mean).

Fig 2. What time scale are you using here for computing the indices? Please, specify.

\*\*\* L133 "in events where both STI and -SPI are jointly exceeded." Not clear what is "jointly exceeded", though this is described rigorously later. At this point, I tended to expect a method that would catch events where STI and -SPI high values are jointly exceeded (e.g., concurrent values above the 99.5th percentile). In fact, the authors also refer to "The highest probability of concurrent hot-dry events" at line 172 and later in the paper, when discussing the results based on the copula-related metric. Is there any particular reason for opting for this particular copula-based threshold criterion?

Selecting (u,v) pairs such that C(u,v)> threshold implies to pick up values of (u,v) which are beyond the "threshold curve" defined by C(u,v)=threshold. Depending on the dependence between -SPI and STI (which depends on the location), the "threshold curve" in the [0,1]x[0,1] space will be different (also the number of selected events will depend on the dependence, which is not something to criticise). Hence, one may wonder whether this leads to comparing events at different locations that are different in nature. Hence, whether using concurrent extreme would not lead a more natural interpretation of the results.

I would appreciate a brief discussion that considers the above, such to provide some insights to the reader.

Hence, in the next, could you find and use a different term than "concurrent hot and dry events"?

L140, do you mean? "For any given time scale, we define the spatial extent of the compound event as the percentage of grid cells affected by the compound event. "

*** L 143-145. This is not fully clear. E.g., "median" among which sample? Therefore I had issues in understanding the results on this topic fully. Please, clarify.

L131-140, Please use the same term when you refer to the same concept to avoid misunderstanding. I got that with "compound hot-dry events", "extreme droughts", and "compound events" you are referring to the same thing in these lines.

Figure 3,
- I assume that the different simulated lines correspond to the 100 simulated samples. Please specify in the caption.
- In b and d, precipitation appear to behave a bit differently from observations. However, this may just be a result of higher variability of the precipitation, compared to temperature. Hence, if there were confidence interval around observations, one may find that both T and P behave similarly in term of overlapping the confidence interval. Please, consider the following: Would it be possible to add some confidence interval of the observation estimates? For the autocorrelation function, adding a line highlighting the level of significant correlation may help.
- Panel e-f should have the same axis to facilitate the comparison.

If the above lead to some changes in the interpretation of the graphs/evaluation, then this should be mentioned in the text. However, overall, given that the aim is to discuss the model performance, I do not think that the text should be too much related to the specific performance at an individual grid point. Rather, try to summarise the characteristics of the model at most grid points (as I guess you did already via selecting a representative grid point).

You could add a few words to the last sentence ("The model is considered suitable for the analysis of compound hot-dry events because it has an acceptable performance with respect to all three aspects.") such to highlight that, despite there are some limitations, your model do at least offer a way to tackle the challenging study of such a compound event.

Figure 4. Missing a full stop before "(b)". You could add, probably in the caption, a brief reference to the variogram, e.g., that describes the degree of spatial dependence of a field (add reference).

L164. I would divide the first sentence in two sentences. The second sentence should highlight (as you already imply) that the maps allow for evaluating the spatial pattern of the indices, rather the magnitude (I guess that the index is computed on observed and simulated sample independently so they provide information the anomalies relative to the climatology in observations, and simulations, respectively).

L176, Figure 6. The author should mention that the simulations tend to underestimate compound hot and dry events. This seems in line with what discussed at line 158 (on the dependence between P and T).

L179 This is an interesting result.
"However, the probability of concurrent events decreases with increasing time scale, as can be expected due to the increasing aggregation of multiple weather events in longer periods…"

I understand that the point is that multiple weather events, not all of which favouring instantaneous concurrent hot and dry conditions, are pulled together at long time scales. Hence, the overall dependence is influenced by a combination of weather events, some of which causing and others not causing dependence. As a result, the dependence is weakened compared to the short term case where dependence-driving weather system are considered individually. You may explain this more explicitly, if you agree with me.

L196, do you mean "importance of T and P"?
*** L196-200 is not clear, see my comment above on the methodology. Please, improve this.

*** L216, This is a finding that could have been found also based on observations only. I am wondering whether the authors could highlight in the discussion the features that the weather generator (e.g., longer time scale) allowed, in this analysis, to understand better than based on observations only.

L219, is this reasoning apply also for the yearly time scale?

L228, consider adding some references.

---

## Author Response (AR1)

Dear Authors,

Both reviews judge you manuscript suitable for publication in ESD after some revisions. In your responses you have already outlined how you would address each of the comments, thank you.

In addition to the reviewer comments, I have one more comment regarding the seasonality of compound hot-dry events. Compound hot-dry events are usually studied in a fixed season, typically the summer, where we have a good understanding on the processes that lead to the dependence between hot and dry events. In contrast, in winter, the dependence might be opposite. Your Fig. 2 is a good example of this behaviour. This distinction may be important as compound hot-dry events in summer have different impacts that compound hot-dry event in winter. In your plots on compound event characteristics (Figs. 5-10), however, all seasons are aggregated and it is difficult to judge how the potentially varying dependencies over the year affect these aggregated statistics. For instance, one might question how informative it is to state the average probability of concurrent events when the dependence strongly varies across season (e.g. Fig. 6). In particular, opposite dependencies between winter and summer might result in (annual) compound event occurrence probabilities that are consistent with the independent case even though their occurrence might by much higher in summer and much lower in winter (I suspect that is happening in many ares in Fig. 6). I therefore would like to ask you to add some analysis/discussion that addresses this aspect in the revised manuscript.

Best regards,

Jakob Zscheischler

*Dear Jakob,*

*Thank you very much for your thorough assessment and for pointing out the need to address compound event seasonality in addition to the points risen by the two reviewers. We agree that the main season of interest when studying compound hot-dry events is summer because of the potential impacts during this season. Our focus on high T (SPI)- low P events (-SPI) ensures that we are only focusing on the upper right tail of the bivariate distribution. I.e. our event extraction procedure, which is based on the empirical copula ensures that we are not selecting events in one of the other tails e.g. low T-low P events. This focus on high T-low P events ensures that we are only selecting events in the summer season. We now address this point in the Methods section of our manuscript by stating: 'Please note that the focus on high T and low P events leads to the selection of compound events in the summer season. For an aggregation period of 1 month, all selected compound events happen between May and October with over 90% of the events happening in July or August. The seasonal focus is slightly shifted towards late summer (august) and early fall (september and october) as we move towards longer aggregation periods.' (see p. 7, lines 162-165).*

*The new version of our manuscript improves and expands the presentation of methodological aspects and complements the discussion section by discussing model limitations in more detail as suggested by the reviewers.*

*Please find our detailed answers to the reviewers' comments in our point-by-point response below. We hope that you find the revised version of our manuscript suitable for publication in ESD.*

*On behalf of all co-authors,*

*Manuela Brunner*

**Reviewer 1**

Review of the manuscript "Space-time dependence of compound hot-dry events in the United States: assessment using a multi-site multi-variable weather generator" by Manuela I. Brunner, Eric Gilleland, and Andrew W. Wood.

**General comment**

The authors introduce a multi-site multi-variable weather generator (*PRSim.weather*), which allows for simulating Temperature and Precipitation over the US during 100*28 years. While the weather generator has some limitations (that the authors discuss), the output is overall satisfying. The simulated data allows for analysing both (1) events that tend to be characterised by hot and dry conditions and (2) the spatial extents of these events. The authors illustrate and discuss the characteristics of these events across the US. Some improvements is needed, especially in the presentation of some methodological aspect (selection of concurrent hot and dry events and method for analysis in Figure 10). Generally, the paper is definitely well structured and I found it interesting. I recommend the authors to consider my specific comments below. Those marked with *** are the less technical.

**Reply:** *We thank the reviewer for their detailed and thoughtful comments, which particularly helped to improve the presentation of some methodological aspects. Please find our responses to the individual comments below.*

**Specific comments**

L9 **meteorological** drought indicators
**Reply:** *We specified that we are referring to meteorological drought indicators.*
**Modification:** p.1, l.9.

L25. Could you mention, very *briefly* as it is an introduction, what were the causes for the changes in hot and dry events in these studies, e.g., temperature/precipitation trends?
**Reply:** *Most of the studies that look at the drivers of changes indicate that increasing temperatures can at least partly explain the changes in hot-dry events. We specify that 'Substantial increases in the number of concurrent droughts and heat waves over the last few decades that are partly*

*explained by increasing temperatures have been reported not just for the US…'*
**Modification:** p.2, l.26.

L30 I suggest re-shaping the sentence slightly. That is, including the words "local" and "regional" (or "aggregated over a region"). The local impact depends on frequency and duration. The aggregated regional impacts depend, in addition, also on the extent.
**Reply:** *We integrated the terms local and regional by writing: 'While frequency of occurrence is an important factor determining local and regional impacts, the severity of impacts related to compound events likely also depends on their spatial extent, i.e. how large the affected region is, and their time scale, i.e. whether they just last weeks or extend over a longer period of time.'*
**Modification:** p.2, l.30-32.

*** L40. I think that the terminology could be improved, not only here, despite it is not wrong as there is no full agreement on this matter in the community. For example, here: "(2) spatial extents of compound events". - You use "compound event" to refer to concurrent events or multivariate events (such as hot-dry events), which is a type of compound event. - Note that the spatial characteristics of an event, make the event compound on its own (Zscheischler et al., 2020). Therefore, the considered events are compound for two reasons, however you refer to the multivariate characteristic as a compound element, but you do not do the same for the spatial part. Of course, you cannot say twice "compound", but why is one privileged? Talking of "spatial extents of concurrent hot and dry events" may make things better in the paper. This would lead to reshaping a bit, for example, lines 40-45.
**Reply:** *Thank you for this note. We are aware that the spatial characteristics of an event make it compound on its own. However, the word compound itself is unspecific as it does not tell us anything about where the 'compoundedness' comes from. In our attempt to specify the nature of compoundedness, we came up with the term 'spatial compound'. We agree though that this term is still not specific enough. We therefore either replaced the term 'spatial compound' by the more specific term 'spatial multivariate' or by the more general term 'compound'. Modifications were applied throughout the document.*
**Modification:** several instances throughout the manuscript

L40. Spatial patterns and spatial extents. Please, make the difference clear. I know what you mean, but I suspect that it will not be obvious to everyone.
**Reply:** *We added the following specifications: '(1) spatial patterns of compound event affectedness (i.e. where in the US hot-dry events are most frequent), (2) spatial extents of compound events (i.e. how large compound events are).'*
**Modification:** p.2, l.40-41.

L45. This statement is interesting. We have recently worked on the topic and shown that it is very difficult to study seasonal precipitation extreme extents without large ensemble simulations (discussed at the end of the "Present-day spatial scale extremes" section): *Bevacqua, E., Shepherd, T.G., Watson, P.A.G., Sparrow, S., Wallom, D., and Mitchell, D. (2020). "Larger spatial footprint of wintertime total precipitation extremes in a warmer climate". Submitted. Preprint's DOI: 10.1002/essoar.10505310.1*

**Reply:** *We agree that using large ensemble simulations would be an alternative to using stochastic models. We therefore slightly adjusted the sentence to: 'This challenge can for example be tackled…'. In the discussion section, we add that: 'If physical consistency is a requirement for a specific application, stochastic approaches may be combined with physical approaches as e.g. in the weather generator AWE-GEN-2 by Peleg et al (2017) or one may rely on large climate ensemble simulation approaches (Deser et al. 2020; Bevaqua et al, 2020).*
**Modification:** p.15, l.243-245.

\*\*\* L45 You write: "Studying such spatial compound events is challenging because they are rare in observational records (Zscheischler et al., 2018). This challenge can be tackled by developing stochastic simulation approaches to generate large data sets with similar statistical properties as the observations "The weather generator is calibrated on and learn from the limited observation (or available data). So, does using a weather generator address completely the challenge of limited data? I suggest discussing this, especially the limitations, for a non-expert reader.
**Reply:** *Yes, as every other calibrated/fitted model, the weather generator is based on the available observations. We specified in the Methods section that: 'The simulation of yet unobserved magnitudes becomes possible thanks to the use of parametric distributions for T and P in Step 2.' We added the following point to the discussion: 'Please note that even though the model generates yet unobserved observations, the simulations are not independent of the limited sample size used to fit the model because the model is data-driven as any other calibrated/fitted model.'*
**Modification:** p.5, l.127-128; p.15, l.239-241.

L76 add "daily" to "time series".
**Reply:** *We added 'daily'.*
**Modification:** p.3, l.71.

L83 Also in the procedure. You simulate, in the end, daily time series of P and T. Could you state this explicitly somewhere, maybe simply adding a "daily" somewhere?
**Reply:** *We specified that PRSim.wave simulates compound hot-dry events 'at a daily scale'.*
**Modification:** p.3, l.85.

Caption Fig 1. Add "daily" and "monthly" where required. E.g., in Step (2), I suggest moving the "monthly": "fit SEP distribution to T and E-GP distribution to P **monthly** time series of all sites"
**Reply:** *We added 'daily' to clarify that both input and output are at a daily scale and we clarified that the SEP and E-GP distributions were fitted 'at a monthly scale'.*
**Modification:** p.6, figure caption 1.

L119 Adding "aggregated" somewhere may help to make very clear that you will pull together all the weather generator output in a unique aggregated time series of 2800 years (one may in principle repeat the analysis on the 100 weather generator output and get, e.g., a mean).
**Reply:** *Thank you for this suggestion. We specified that: '…increase the sample size available for the assessment of compound hot-dry events by pooling the different model runs.'*
**Modification:** p.5, l.137.

Fig 2. What time scale are you using here for computing the indices? Please, specify.
**Reply:** *We specified that Figure 2 shows 'monthly' time series.*
**Modification:** p.7, figure caption 2

*** L133 "in events where both STI and -SPI are jointly exceeded." Not clear what is "jointly exceeded", though this is described rigorously later. At this point, I tended to expect a method that would catch events where STI and -SPI high values are jointly exceeded (e.g., concurrent values above the 99.5th percentile). In fact, the authors also refer to "The highest probability of concurrent hot-dry events" at line 172 and later in the paper, when discussing the results based on the copula-related metric. Is there any particular reason for opting for this particular copula-based threshold criterion? Selecting (u,v) pairs such that C(u,v)> threshold implies to pick up values of (u,v) which are beyond the "threshold curve" defined by C(u,v)=threshold. Depending on the dependence between -SPI and STI (which depends on the location), the "threshold curve" in the [0,1]x[0,1] space will be different (also the number of selected events will depend on the dependence, which is not something to criticise). Hence, one may wonder whether this leads to comparing events at different locations that are different in nature. Hence, whether using concurrent extreme would not lead a more natural interpretation of the results. I would appreciate a brief discussion that considers the above, such to provide some insights to the reader. Hence, in the next, could you find and use a different term than "concurrent hot and dry events"?
**Reply:** *Thank you for highlighting that the definition of compound hot-dry events needed further explanation. We applied the threshold to the bivariate distribution of T and P instead of their marginal distributions, which is one potential way of identifying bivariate extremes. A joint definition where both T and P have to exceed a marginal threshold will extract only events in the upper right corner of the distribution. Using a threshold on the bivariate distribution also includes these upper right corner events and adds a few events which are also critical but only extreme with respect to one of the margins. A nice illustration of the different probability spaces we are talking about is given in Figure 1 of Serinaldi et al. (2015) (https://link.springer.com/article/10.1007/s00477-014-0916-1). We agree that talking about 'joint exceedances' and about 'concurrence' is confusing in this context and therefore replaced the terms by more appropriate ones (concurrence = compound). We also added the following specification to the methods section: 'This copula-based threshold procedure slightly differs from an approach where both margins (-SPI and STI) have to jointly exceed a threshold in order for an event to be defined as a compound event. The bivariate threshold procedure includes a slightly bigger event space, which besides the jointly marginally extreme events also includes those events that are extreme in terms of the bivariate distribution but not necessarily in terms of both margins.'*
**Modification:** p.7, l.158-162.

L140, do you mean? "For any given time scale, we define the spatial extent of the compound event as the percentage of grid cells affected by the compound event. "
**Reply:** *Thank you for this rephrasing suggestion, which we integrated as: 'To assess the spatial extent of compound events at different time scales, we define the spatial extent of the compound event as the percentage of grid cells affected by the compound event at any given time scale.'*
**Modification:** p.7, l.166-167

*** L 143-145. This is not fully clear. E.g., "median" among which sample? Therefore I had issues in understanding the results on this topic fully. Please, clarify.

**Reply:** *Thank you for pointing out the need for clarification. We specified that: 'we compute Kendall's correlation between the median bivariate distribution (empirical copula) and the median standardized indices STI and SPI over all simulation runs at different time scales.' We further clarify that 'This correlation analysis is performed for nine hydro-climatic regions in the United States to quantify the regional spread in the role of STI and SPI for compound event development, i.e., correlation is computed between median bivariate distributions and median STI or SPI at different grid cells within a region.'*

**Modification:** p.7, l.172-173.

L131-140, Please use the same term when you refer to the same concept to avoid misunderstanding. I got that with "compound hot-dry events", "extreme droughts", and "compound events" you are referring to the same thing in these lines.

**Reply:** *The term 'extreme droughts' was indeed confusing. We replaced it with 'compound event' to use consistent terminology.*

**Modification:** p.7, l.157.

Figure 3, - I assume that the different simulated lines correspond to the 100 simulated samples. Please specify in the caption. - In b and d, precipitation appear to behave a bit differently from observations. However, this may just be a result of higher variability of the precipitation, compared to temperature. Hence, if there were confidence interval around observations, one may find that both T and P behave similarly in term of overlapping the confidence interval. Please, consider the following: Would it be possible to add some confidence interval of the observation estimates? For the autocorrelation function, adding a line highlighting the level of significant correlation may help. - Panel e-f should have the same axis to facilitate the comparison. If the above lead to some changes in the interpretation of the graphs/evaluation, then this should be mentioned in the text. However, overall, given that the aim is to discuss the model performance, I do not think that the text should be too much related to the specific performance at an individual grid point. Rather, try to summarise the characteristics of the model at most grid points (as I guess you did already via selecting a representative grid point). You could add a few words to the last sentence ("The model is considered suitable for the analysis of compound hot-dry events because it has an acceptable performance with respect to all three aspects.") such to highlight that, despite there are some limitations, your model do at least offer a way to tackle the challenging study of such a compound event.

**Reply:** *We specified that the different lines in a-d refer to the different simulation runs. Adding confidence intervals to the observations would be possible by applying some bootstrap procedure. However, we think that the observed and simulated acfs appear to be similar enough not necessarily requiring this additional uncertainty information. Panels e and f are displayed on the same scale, which is indeed important to facilitate comparison. However, this might not have been sufficiently clear because there are two separate y-axes, one for T and one for P. We therefore specified in the caption that the left y-axis refers to precipitation and the right axis to temperature. We here present a model evaluation for the local T and P characteristics for one single grid cell, which was not*

*sufficiently clear in the previous version of the manuscript. We specified that local model performance was assessed on an example grid cell and highlight that 'The above-described model evaluation can be generalized to other grid cells in the data set.' We also slightly adjusted the last sentence of the paragraph to highlight the main benefit of the stochastic simulation approach, i.e. increasing the available sample size of compound events: 'The model is considered suitable for the analysis of compound hot-dry events because it has an acceptable performance with respect to all three aspects and enables increasing the sample size of compound events.'*
**Modification:** p.9, figure caption 3; p.8, l.187; p.8, l.192.

Figure 4. Missing a full stop before "(b)". You could add, probably in the caption, a brief reference to the variogram, e.g., that describes the degree of spatial dependence of a field (add reference).
**Reply:** *We added an 'and' to separate the descriptions of subpanels (a) and (b). We also added a short description of a variogram including a suitable reference: 'which describe the degree of spatial dependence of a field (Cressie, 1993).'*
**Modification:** p.9, figure caption 4.

L164. I would divide the first sentence in two sentences. The second sentence should highlight (as you already imply) that the maps allow for evaluating the spatial pattern of the indices, rather the magnitude (I guess that the index is computed on observed and simulated sample independently so they provide information the anomalies relative to the climatology in observations, and simulations, respectively).
**Reply:** *We split up the sentence into two sentences with a second sentence stating: 'These spatial samples enable comparing observed and simulated STI and SPI patterns for different levels of extremeness'. Yes, the indices are computed for observations and simulations independently.*
**Modification:** p.9, l.194-196.

L176, Figure 6. The author should mention that the simulations tend to underestimate compound hot and dry events. This seems in line with what discussed at line 158 (on the dependence between P and T).
**Reply:** *We specified that: 'The spatial STI and SPI patterns are reflected in the spatial distribution of the probability of compound hot-dry events, which is also realistically represented but slightly underestimated by PRSim.weather.'*
**Modification:** p.10, l.201-203.

L179 This is an interesting result. "However, the probability of concurrent events decreases with increasing time scale, as can be expected due to the increasing aggregation of multiple weather events in longer periods…" I understand that the point is that multiple weather events, not all of which favouring instantaneous concurrent hot and dry conditions, are pulled together at long time scales. Hence, the overall dependence is influenced by a combination of weather events, some of which causing and others not causing dependence. As a result, the dependence is weakened compared to the short term case where dependence-driving weather system are considered individually. You may explain this more explicitly, if you agree with me.
**Reply:** *Yes, we definitely agree. We slightly expanded the sentence by writing: 'However, the probability of compound events decreases with increasing time scale, as can be expected due to the*

*aggregation over increasingly longer periods of multiple weather events that may not all favor instantaneous compound hot and dry conditions, and event extremeness.'*
**Modification:** p.11, l.209-211.

L196, do you mean "importance of T and P"?
**Reply:** *Yes, we changed the sentence to 'importance of T (STI) and P (SPI)'.*
**Modification:** p.14, l.226.

\*\*\* L196-200 is not clear, see my comment above on the methodology. Please, improve this.
**Reply:** *In addition to the changes already applied to the methods section, we provided more guidance for reading Figure 10 in the results section: 'T is a particularly important driver at short time scales as indicated by the high correlation between median STI and median bivariate distribution of grid cells within a specific hydro-climatic region' and adjust the Figure caption: 'Importance of T and P as drivers of compound events across time scales and extremeness levels. Correlation of median bivariate distribution (empirical copula) with (a) STI and (b) SPI across simulation runs between grid cells in nine hydro-climatic (Bukovsky) regions (spread of boxplot) per time scale (color) and level of extremeness (hue).'*
**Modification:** p.14, l.227-228; figure caption 10.

\*\*\* L216, This is a finding that could have been found also based on observations only. I am wondering whether the authors could highlight in the discussion the features that the weather generator (e.g., longer time scale) allowed, in this analysis, to understand better than based on observations only.
**Reply:** *It is true that spatial patterns of compound hot-dry events could also have been studied using observations only. However, studying rare spatial events with large extents would have been difficult/impossible. We added the following point to the discussion section: 'It [the stochastic model] enables studying rare spatial multivariate events, which would not be possible using observations only.'*
**Modification:** p.15, l.238-239.

L219, is this reasoning apply also for the yearly time scale?
**Reply:** *No. Spatial variations disappear at annual time scales. We specified that this paragraph refers to the findings for 'sub-annual time scales'.*
**Modification:** p.15, l.257-258

L228, consider adding some references.
**Reply:** *We added a suitable reference to the statement.*
**Modification:** p.16, l.270.

**Reviewer 2**

**Summary:**

In this paper, the authors introduce a multi-site multi-variable stochastic weather generator called "PRSim.weather" to assess the (joint) occurrence probabilities, severity, and spatial patterns of compound hot-dry events in the US at various time scales (1 week, 1 month, 3 months, 6 months, 1 year). The proposed weather generator is a simple extension of a previously published version for a single variable, and they here make some necessary adjustments for its application to study high temperatures / low precipitation. The authors conclude that their model correctly replicates the distribution and dependencies in observed data, and their analysis further reveals that

(1) Northwestern/Southeastern US are more likely to experience hot-dry events

(2) the time scale influences the size of compound hot-dry events (i.e., shorter time scales imply larger spatial extents of joint extreme events)

(3) temperature mostly determines compound events for short time scales, while precipitation is the key factor for longer time scales.

**Assessment:**

Overall I like the paper and the data analysis. The topic tackled by the authors, namely to understand the spatio-temporal distribution of compound extreme weather events, is difficult and timely. The paper is well written, is relatively concise and the authors precisely detail the findings of their analysis. The proposed approach (PRSim.weather) has, however, some limitations that the authors should, I think, better acknowledge and discuss more openly. I discuss some of those in my comments below. Another point to mention is that although the data analysis and the findings are well supported and of practical interest, the methodological novelty is rather limited, since the proposed method is a simple adjustment of an already published approach.

**Reply:** *We thank the reviewer for highlighting the value of our work and for indicating the need to discuss the limitations of the stochastic simulation approach in more detail. We added methodological clarifications where suggested and expanded the discussion section by discussing model limitations in more detail.*

**Comments:**

1. I found the 5 steps of the method on page 4 (lines 90-109) difficult to understand. For example,
**Reply:** *Thank you for highlighting the need for methodological clarification. The methods section was substantially expanded.*

  - how do you "fit monthly distributions to T and P"? do you first a distribution to the data within each month separately assuming that they are iid during that month?

**Reply:** *Yes, we specified that: '(i.e. one separate distribution is fitted to the data in each month)'.*
**Modification:** p.4, l.93-94.

- what do you mean by "we combine the E-GPD with as many zero-values as in the observations"? Do you mean that you don't simulate zero observations, but keep them fixed like in the data? If so, is this not "cheating" (i.e., over-fitting)? and do you keep the zeros at the same time points?
**Reply:** *Yes, the E-GPD is only used to simulate the non-zero part of the distribution similar to most existing stochastic precipitation simulation approaches. However, the zero-values are not pinned to the same time points as in the observations to enable temporally varying precipitation patterns. This reordering is achieved thanks to the rank-ordering in Step 5. We rephrase the description for clarification: 'We use the E-GPD to simulate non-zero precipitation values and complement it with as many zero-values as in the observations to obtain the full P distribution with appropriate probability of precipitation occurrence.'*
**Modification:** p.4, l.109-111.

- how do you apply the continuous wavelet transform? and how to interpret the amplitude and phase signals?
**Reply:** *The continuous wavelet transform was performed using the Morlet mother wavelet and the R-package wavScalogram (function cwt_wst). A reference and the equation of the continuous wavelet transform and the Morlet wavelet were added to the manuscript. The amplitude tells us about the strength of variability at different time scales while the phase tells us about the time shift in the data.*
**Modification:** p.4, l.114-119.

- in point 4., what do you generate a random time series for both T and P? Or just one time series?
**Reply:** *We specified that we generate one random time series based on the temperature time series.*
**Modification:** p.5, l.120.

- in point 5., how to you do the "rank-transform" exactly? Do you mean that you apply the probability integral transform?
**Reply:** *Yes, the rank-transformation is achieved by applying the probability integral transform, which we now specified in the text.*
**Modification:** p.5, l.124-125.

Bottomline: I think it is needed to clarify the methodology. It seems necessary to me to provide further mathematical equations to clarify each point and to illustrate the wavelet transform with a simple example in order to faciliate interpretation.
**Reply:** *We added mathematical equations to explain the SEP and E-GPD distributions, the continuous wavelet transform and the Morlet wavelet. A schematic illustration of the procedure is provided in Figure 1.*
**Modification:** p.4, equations 1 to 4.

2. The methodology seems to have certain limitations that may be concerning:

- The authors mention that the same random phases are used at all sites and for both variables. Is this not too restrictive, and will this not create too strong spatial or cross-dependencies?

**Reply:** *Using the same phases across stations and variables allows us to model spatial and variable dependencies. Without following this procedure, we would produce local simulations, not retaining the spatial dependencies we would like to reproduce. An example of what happens to spatial dependencies if non-identical phases are used across stations is provided in Figure A2 in Brunner and Gilleland (2020; https://hess.copernicus.org/articles/24/3967/2020/). While this step is essential to model spatial and variable dependencies, it is true that neither spatial nor variable dependencies are perfectly represented. In the case of spatial temperature dependencies, we see a slight overestimation (Figure 4) while T-P dependencies are slightly underestimated (Figure 3g-h). Jointly modeling temporal, spatial, and variable dependencies is very challenging and we therefore consider model performance to be satisfactory for our application. We acknowledge these model limitations in the methods section and added an additional statement to the discussion section: 'However, spatial dependencies are slightly overestimated while variable dependencies are slightly underestimated. The model still has acceptable performance across three types of dependencies - temporal, spatial, and variable - and enables studying rare spatial multivariate events, which would not be possible using observations only.'*
**Modification:** p.15, l.236-239.

  - In point 4., a time series of one site is chosen at random. Are all sites "exchangeable"? What is the implication of this approach?
**Reply:** *Yes, the stations are exchangeable as the goal is to generate a random time series with some seasonality. Using a totally random series, e.g. white noise, won't allow us to reproduce temperature seasonality, which is why we are using a random series mimicking the temperature signal.*

  - Again in point 4., a random time series is generated by bootstrap by resampling years with replacement. This implies that years are exchangeable and therefore that any time trend is ignored. Is this not a major issue for temperatures (and perhaps also precipitation)? If so, this should be further acknowledged and discussed.
**Reply:** *It is correct that PRSim.weather is a stationary model, i.e. potential time trends are not considered. This is not an issue in this study as we do not aim to look at temporal trends in compound event characteristics. Adapting the model to non-stationary conditions would primarily require the introduction of non-stationary distributions for P and T in Step 2 of the modeling procedure. If one would in addition want to consider potential non-stationarities in spatial and/or variable dependencies, one would have to use alternative resampling schemes in Step 4 retaining the temporal order of the original series. We add a short paragraph to the discussion stating that 'Extending model application to non-stationary conditions would require the implementation of non-stationary distributions for both T and P. For example, one could introduce covariates for certain parameters of the marginal distributions of T and P in Step 2 or introduce covariates with information about trends or variability in P and/or T to guide resampling in Step 4.'*
**Modification:** p.15, l.253-256

  - Using a bootstrap-based approach implies implicitly that simulated events will NEVER be more extreme than what has been observed in the data. This is a major limitation since the goal here is to enrich the dataset with more simulations of compound extreme events.
**Reply:** *This statement is true for classical bootstrap approaches. However, PRSim.weather is a semi-parametric model, which combines a non-parametric bootstrap model to represent spatial and*

*variable dependencies with two parametric models for temperature and precipitation. This may not have been clear in the previous version of the manuscript and we added that: 'Using theoretical instead of empirical distributions will allow us to generate extreme values more extreme than the observations.'*

**Modification:** p.4, l. 93-94.

- Estimating a copula using the empirical copula (based on ranks) implicitly implies that the data are stationary over time, thus without time trend (or seasonality) again. Is this a reasonable assumption here?

**Reply:** *The STI and SPI time series do not show a time trend in most grid cells and we think that using an empirical copula is appropriate especially because it is a non-parametric model.*

3. L129, p5, "site-specific Gamma distribution": should this not be the E-GPD distribution as specified in the methods section (point 2.)?

**Reply:** *Yes, we corrected this by replacing 'Gamma distribution' with 'E-GPD' distribution.*

**Modification:** p.5, l.147.

4. p6, top: further details on copulas are required to introduce the notation properly...

- What is a copula => Joint distribution with uniform Unif(0,1) margins

- What is C(u,v)? => the copula of T and -P

- What are the ranks R_i and S_i? => ranks of T or P values across the time series

- In Figure 2, what does "Empirical copula" mean? => the values of C_n(R_i/(n+1),S_i/(n+1)), i.e., the empirical copula evaluated at the observed uniform values.

**Reply:** *We clarified the notation as suggested.*

**Modification:** p.6, 150-154; figure caption 2.

5. In Figure 3, the results are almost too good to be true in my opinion. Does this not hide some issues of overfitting? Again, how do you simulate the zeros in precipitation for example?

**Reply:** *We use a four-parameter distribution to fit temperature and a three-parameter distribution to fit precipitation. These distributions are flexible enough to reproduce the main distributional characteristics of P and T. Non-zero precipitation values are added separately as often the case with precipitation distributions, e.g. when combining a Markov Process with a parametric distribution. One could use less flexible distributions with less parameters, which would decrease simulation performance.*

6. When the goal is to simulate many more compound events, it is crucial to check if the marginal and joint tails are captured correctly. For marginal tails, I would suggest to consider comparing long-term return levels of simulated vs observed data (on a scale that zoom into the tail rather than the bulk). For joint tails, a possibility is to look at the tail correlation coefficient ($\lambda(u) = P(U1>u \mid U2>u)$) for increasing thresholds u=0.8,0.9,0.95,0.98,0.99,0.995,0.999, say. Such diagnostics would complement the results in Figure 3.

**Reply:** *Thank you for these suggestions. We estimate the 100-year return levels for T and P and all*

*grid cells for both observed and simulated series. The comparison of observed and simulated return levels shows that observed and simulated return levels estimated using the SEP for temperature and the E-GPD for precipitation are very similar (Figure 1 in this response to the reviewers). This additional analysis confirms that the SEP and E-GPD are indeed good choices to model T and P, respectively. We also compute upper tail dependence for different thresholds for high T and low P values. The tail dependence between extremely low P and high T is 0. The simulations reproduce this behavior.*

[Figure]

*Figure 1: Observed vs. simulated 100-year return levels for (a) temperature (°) and (b) precipitation (mm/d).*

7. In Figure 4, the simulated fields appear smoother than observations. Why is that the case?
**Reply:** *In the case of temperature, this is indeed true. This slight overestimation in spatial dependence possibly comes from the phase randomization procedure which relies on random phases generated from bootstrapped temperature time series. As mentioned earlier, we added this point to the discussion.*
**Modification:** p.15, l.236-238.

8. In Figure 5, it seems like the spatial extent of very extreme events is largely overestimated. Is this because a single random phase is chosen across sites? Or is this a false impression due to the fact that there are less extreme events available in observations than simulations?
**Reply:** *Figure 5 maps median observed and simulated STIs and SPIs at a grid scale. It therefore shows that simulated STIs and SPIs are more extreme than observed ones. The simulated medians are more extreme because the model is able to generate more extreme events than in the observations thanks to the theoretical distributions used to simulate T and P distributions. We specified in the figure caption that the median events refer to 'a certain grid cell' and that 'While the simulated spatial STI and SPI patterns look similar as the observed ones, they are more expressed because of the larger sample available, which contains yet unobserved extremes because of the use of parametric distributions for simulating T and P'.*
**Modification:** p.10, figure caption 5; p.10, l.195-197.

9. In Figure 6, simulations severley underestimate the joint probability of concurrent events for severe and extreme events... and also for moderate events in the Southeastern part of the US... Is this due to using the empirical copula approach? What is the cause of this and how to remediate

this (fairly severe) issue?

**Reply:** *The underestimation of the co-occurrence probability of compound events is related to the underestimation of T-P dependence as illustrated in Figure 3, acknowledged in the Methods section and discussed in the Discussions section. The reduction in variable dependence is introduced in the backtransformation step, which can hardly be avoided (Embrechts et al. 2002; Correlation and dependence in risk management: Properties and pitfalls). A potential improvement of the representation of variable dependence may be achieved by using phase annealing, which modifies the phases in an iterative way in order to optimize certain statistics but increases the computational effort (Hoerning et al. 2018; Phase annealing for the conditional simulation of spatial random fields).*

10. Figure 8 plots the "median spatial extent of concurrent events affecting grid cell". How was that calculated? I don't think it is clearly explained in the text...

**Reply:** *Thank you for pointing out the need for clarification. We specified in the Methods section that: 'To assess the spatial extent of compound events at different time scales, we define the spatial extent of the compound event as the percentage of grid cells affected by the compound event at any given time scale. Then, for each grid cell, we determine the median spatial extent of those events it is affected by.'*

**Modification:** p.7, l.166-168.

11. Figure 10 reports the values of Kendall's tau between T and the bivariate empirical copula, as well as between P and the bivariate empirical copula. However, given that the empirical copula is itself calculated from T and P, I'm not convinced that such "correlation" values make sense... Wouldn't it make more sense to report the actual ranks $R_i/(n+1)$ and $S_i/(n+1)$, which already give the importance of T and P in the calculation of the empirical copula?

**Reply:** *With this part of the analysis, we intend to explain which of the two variables is related most strongly to the empirical copula, i.e. represents the main driver of the compound event. Reporting just the ranks would not allow us to provide a measure of association and ranks for both variables would range from 1 to n.*

---

## Referee Report (RR1)

Review of the manuscript "Space-time dependence of compound hot-dry events in the United States: assessment using a multi-site multi-variable weather generator" by Manuela I. Brunner, Eric Gilleland, and Andrew W. Wood.

I thank the authors for considering my comments. They addressed my comments well and I find the paper improved. Therefore I recommend to publish the manuscript after some final technical revisions that can be applied based on my few last comments below.

**Comments**

1) L30 I suggest re-shaping the sentence slightly. That is, including the words "local" and "regional" (or"aggregated over a region"). The local impact depends on frequency and duration. The aggregated regional impacts depend, in addition, also on the extent.
Reply: We integrated the terms local and regional by writing: 'While frequency of occurrence is an important factor determining local and regional impacts, the severity of impacts related to compound events likely also depends on their spatial extent, i.e. how large the affected region is, and their time scale, i.e. whether they just last weeks or extend over a longer period of time.'

Here there was misunderstanding. I would delete "and regional" or something along this line. "Regional" already implies a spatial aggregation, which is something that comes with the spatial extent later on in the sentence.

2) L45. This statement is interesting. We have recently worked on the topic and shown that it is very difficult to study seasonal precipitation extreme extents without large ensemble simulations (discussed at the end of the "Present-day spatial scale extremes" section): Bevacqua, E., Shepherd, T.G., Watson, P.A.G., Sparrow, S., Wallom, D., and Mitchell, D. (2020). "Larger spatial footprint of wintertime total precipitation extremes in a warmer climate". Submitted. Preprint's DOI: 10.1002/essoar.10505310.1
Reply: We agree that using large ensemble simulations would be an alternative to using stochastic models. We therefore slightly adjusted the sentence to: 'This challenge can for example be tackled…'. In the discussion section, we add that: 'If physical consistency is a requirement for a specific application, stochastic approaches may be combined with physical approaches as e.g. in the weather generator AWE-GEN-2 by Peleg et al (2017) or one may rely on large climate ensemble simulation approaches (Deser et al. 2020; Bevaqua et al, 2020).

The sentence "If physical consistency is a requirement" unnecessarily weakens your study. I suggest something along the lines of the text I drafted below. Note that the second sentence goes in the direction of the next comment I had provided in my original review document. The text is related to the fact that the weather generator, being based on observations, may have limit in simulating events that strongly differ from the observed events.
"We note that stochastic approaches may be combined with physical approaches as e.g. in the weather generator AWE-GEN-2 by Peleg et al (2017). In addition, large climate ensemble simulation approaches can allow for gaining information on yet unseen events that may be particularly different in nature from the observed events and therefore may not be simulated by an observation-driven weather generator (Deser et al. 2020; Bevacqua et al, 2020)."

Bevaqua is spelled wrongly. It is Bevacqua et al. has a new citation:
Bevacqua, E., Shepherd, T.G., Watson, P.A.G., Sparrow, S., Wallom, D., and Mitchell, D. (2021). "Larger spatial footprint of wintertime total precipitation extremes in a warmer climate". Geophysical Research Letters, DOI: 10.1029/2020GL091990

3) On the change you applied to describe the copula-based criterion to select compound events:
Note that "bigger" is likely incorrect, as the difference between the size of the two spaces depends on the thresholds used to define the two spaces.

4) L168, "…at any given time scale. Then, for each grid cell, we determine the median spatial extent of those events it is affected by. "
"at any given time scale" should be also in the second sentence to make clear you are not mixing results from different time scales through the median.

5) Please, revise L171 and the caption of figure 10 carefully. I understand what you do, but it was not straightforward to get the point right.

For example, "across simulation" may sometime (also in the caption of figure 10) be misunderstood as a correlation between values of the different 100 runs.

About the text: L170 Text: "To explain the role of the individual variables T and P in compound event occurrence, we compute Kendall's correlation between the median bivariate distribution (empirical copula) and the median standardized indices STI and SPI over all simulation runs at different time scales. This correlation analysis is performed for nine hydro-climatic regions in the United States (Bukovsky; Bukovsky, 2011) to quantify the regional spread in the role of STI and SPI for compound event development, i.e., correlation is computed between median bivariate distributions and median STI or SPI at different grid cells within a region."

I suggest revising the text making sure that all the following steps are clear to the reader. For a given time scale of interest:
- **Divide in regions**
- At each location in the region, compute the median values **based on the time series.**
- Compute the correlation **across** the medians at the different locations in the region
- Present the region-based correlations in **box plots to show the spread of the relationship**

Regarding the results of the correlation itself, some guidance on the interpretation would certainly help the reader and is welcome.

L229, I would remove "occurrence" as your correlation between median values does not account for occurrence, rather it provides indirect information on the occurrence.

Best regards.

---

## Author Response (AR2)

**Reviewer 1**

- Equation (1): please mention what is \gamma(.)?
**Reply:** *\gamma(.) represents the upper tail of the incomplete gamma function, which was specified in the text.*
**Modification:** p.4, l.99-100

- Equation (3): shouldn't it be \phi_0^*? (is the subscript 0 missing?)
**Reply:** *Indeed, the subscript was missing, which we corrected in equation 3.*
**Modification:** p.4, equation 4

- I suggest that the authors report their results regarding the 100-year return levels and the tail dependence coefficient in the main paper.
**Reply:** *Thank you for this suggestion. We added the following sentence to our model evaluation: 'The suitability of the SEP and E-GPD distributions to model local T and P distributions also extends to the tails as 100-year return levels estimated from the observed and simulated series compare well for both variables.'*
**Modification:** p.8, 182-184

**Reviewer 2**

I thank the authors for considering my comments. They addressed my comments well and I find the paper improved. Therefore I recommend to publish the manuscript after some final technical revisions that can be applied based on my few last comments below.

Comments

1) L30 I suggest re-shaping the sentence slightly. That is, including the words "local" and "regional" (or"aggregated over a region"). The local impact depends on frequency and duration. The aggregated regional impacts depend, in addition, also on the extent. Reply: We integrated the terms local and regional by writing: 'While frequency of occurrence is an important factor determining local and regional impacts, the severity of impacts related to compound events likely also depends on their spatial extent, i.e. how large the affected region is, and their time scale, i.e. whether they just last weeks or extend over a longer period of time.'

Here there was misunderstanding. I would delete "and regional" or something along this line. "Regional" already implies a spatial aggregation, which is something that comes with the spatial extent later on in the sentence.
**Reply:** *We removed both 'local and regional' because we agree that frequency and occurrence are generally important determinants of impact independent of whether the event is local or regional. The new sentence reads as follows: 'While frequency of occurrence is an important factor determining impacts, the severity of impacts related to compound events likely also depends on their spatial extent, i.e. how large the affected region is, and their time scale, i.e. whether they just last weeks or extend over a longer period of time.'*
**Modification:** p.2, l.30-34

2) L45. This statement is interesting. We have recently worked on the topic and shown that it is very difficult to study seasonal precipitation extreme extents without large ensemble simulations (discussed at the end of the "Present-day spatial scale extremes" section): Bevacqua, E., Shepherd, T.G., Watson, P.A.G., Sparrow, S., Wallom, D., and Mitchell, D. (2020). "Larger spatial footprint of wintertime total precipitation extremes in a warmer climate". Submitted. Preprint's DOI: 10.1002/essoar.10505310.1
Reply: We agree that using large ensemble simulations would be an alternative to using stochastic models. We therefore slightly adjusted the sentence to: 'This challenge can for example be tackled...'. In the discussion section, we add that: 'If physical consistency is a requirement for a specific application, stochastic approaches may be combined with physical approaches as e.g. in the weather generator AWE-GEN-2 by Peleg et al (2017) or one may rely on large climate ensemble simulation approaches (Deser et al. 2020; Bevaqua et al, 2020).

The sentence "If physical consistency is a requirement" unnecessarily weakens your study. I suggest something along the lines of the text I drafted below. Note that the second sentence goes in the direction of the next comment I had provided in my original review document. The text is related to the fact that the weather generator, being based on observations, may have limit in simulating events that strongly differ from the observed events. "We note that stochastic approaches may be combined with physical approaches as e.g. in the weather generator AWE-GEN-2 by Peleg et al (2017). In addition, large climate ensemble simulation approaches can allow for gaining information on yet unseen events that may be particularly different in nature from the observed events and therefore may not be simulated by an observation-driven weather generator (Deser et al. 2020; Bevacqua et al, 2020)."
**Reply:** *Thank you for these rephrasing suggestions, which we partly adopted: 'We note that stochastic approaches may be combined with physical approaches as e.g. in the weather generator AWE-GEN-2 by (Peleg et al. 2017) or one may rely on large climate ensemble simulation approaches (Deser et al. 2020 and Bevacqua et al. 2021).' Because also observation-driven weather generators such as the one presented in this study can generate yet unseen events, we did not adopt the second sentence proposed.*
**Modification:** p.15, l. 246-248

Bevaqua is spelled wrongly. It is Bevacqua et al. has a new citation: Bevacqua, E., Shepherd, T.G., Watson, P.A.G., Sparrow, S., Wallom, D., and Mitchell, D. (2021). "Larger spatial footprint of wintertime total precipitation extremes in a warmer climate". Geophysical Research Letters, DOI: 10.1029/2020GL091990
**Reply:** *We adjusted the reference accordingly.*
**Modification:** p.18: l. 334-336

3) On the change you applied to describe the copula-based criterion to select compound events: Note that "bigger" is likely incorrect, as the difference between the size of the two spaces depends on the thresholds used to define the two spaces.
**Reply:** *We replaced 'bigger' by 'different'. However, if the same percentile is chosen to define both spaces, bigger should be correct.*
**Modification:** p.7, l. 161

4) L168, "...at any given time scale. Then, for each grid cell, we determine the median spatial extent of those events it is affected by. " "at any given time scale" should be also in the second sentence to make clear you are not mixing results from different time scales through the median.
**Reply:** *Thank you for this suggestion. We included time scale in the second sentence as well: 'Then, for*

*each grid cell, we determine the median spatial extent of those events it is affected by at each time scale.'*

**Modification:** p.7, l.169

5) Please, revise L171 and the caption of figure 10 carefully. I understand what you do, but it was not straightforward to get the point right. For example, "across simulation" may sometime (also in the caption of figure 10) be misunderstood as a correlation between values of the different 100 runs.

About the text: L170 Text: "To explain the role of the individual variables T and P in compound event occurrence, we compute Kendall's correlation between the median bivariate distribution (empirical copula) and the median standardized indices STI and SPI over all simulation runs at different time scales. This correlation analysis is performed for nine hydro-climatic regions in the United States (Bukovsky; Bukovsky, 2011) to quantify the regional spread in the role of STI and SPI for compound event development, i.e., correlation is computed between median bivariate distributions and median STI or SPI at different grid cells within a region."

I suggest revising the text making sure that all the following steps are clear to the reader. For a given time scale of interest:

- Divide in regions
- At each location in the region, compute the median values based on the time series.
- Compute the correlation across the medians at the different locations in the region
- Present the region-based correlations in box plots to show the spread of the relationship

Regarding the results of the correlation itself, some guidance on the interpretation would certainly help the reader and is welcome.

**Reply:** *We agree that the phrasing in the caption was indeed not ideal. We rephrased to: 'Correlation of median bivariate distribution (empirical copula) per grid cell with median (a) STI and (b) SPI per grid cell cell. Correlations were computed using all simulation runs for nine hydro-climatic (Bukovsky) regions (spread of boxplot) per time scale (color) and level of extremeness (hue).' This version covers the four aspects highlighted above.*

**Modification:** p.14, caption Figure 10

L229, I would remove "occurrence" as your correlation between median values does not account for occurrence, rather it provides indirect information on the occurrence.

**Reply:** *We removed the word 'occurrence'.*

**Modification:** p.14, l.229